# Successful Ultrasound-Guided Methotrexate Intervention in the Treatment of Heterotopic Interstitial Pregnancy: A Case Report and Literature Review

**DOI:** 10.3390/jpm13020332

**Published:** 2023-02-15

**Authors:** Ping Li, Xiao Tan, Yi Chen, Qiaoli Ge, Haiying Zhou, Renrong Zhang, Yue Wang, Min Xue, Ruifang Wu, Desheng Sun

**Affiliations:** 1Department of Ultrasonography, Peking University Shenzhen Hospital, Shenzhen 518036, China; 2Department of Ultrasonography, Weifang People’s Hospital, Weifang 261044, China; 3Department of Zhuhai Campus, Zunyi Medical University, Zunyi 563006, China; 4Center of Obstetrics and Gynecology, Peking University Shenzhen Hospital, Shenzhen 518036, China; 5Institute of Obstetrics and Gynecology, Shenzhen PKU-HKUST Medical Center, Shenzhen 518036, China

**Keywords:** heterotopic interstitial pregnancy, in vitro fertilization embryo transfer, interventional ultrasound

## Abstract

Purpose: This study aims to share the experience of minimally invasive ultrasound-guided methotrexate intervention in the treatment of heterotopic interstitial pregnancy (HIP) with good pregnancy outcomes, and to review the treatment, pregnancy outcomes, and impact on the future fertility of HIP patients. Methods: The paper describes the medical history, clinical manifestations, treatment history, and clinical prognosis of a 31-year-old woman with HIP, and reviews cases of HIP from 1992 to 2021 published in the PubMed database. Results: The patient was diagnosed with HIP by transvaginal ultrasound (TVUS) at 8 weeks after assisted reproductive technology. The interstitial gestational sac was inactivated by ultrasound-guided methotrexate injection. The intrauterine pregnancy was successfully delivered at 38 weeks of gestation. Twenty-five HIP cases in 24 studies published on PubMed from 1992 to 2021 were reviewed. Combined with our case, there were 26 cases in total. According to these studies, 84.6% (22/26) of these cases were conceived by in vitro fertilization embryo transfer, 57.7% (15/26) had tubal disorders, and 23.1% (6/26) had a history of ectopic pregnancy; 53.8% (14/26) of the patients presented with abdominal pain and 19.2% (5/26) had vaginal bleeding. All cases were confirmed by TVUS. In total, 76.9% (20/26) of intrauterine pregnancies had a good prognosis (surgery vs. ultrasound interventional therapy 1:1). All fetuses were born without abnormalities. Conclusions: The diagnosis and treatment of HIP remain challenging. Diagnosis mainly relies on TVUS. Interventional ultrasound therapy and surgery are equally safe and effective. Early treatment of concomitant heterotopic pregnancy is associated with high survival of the intrauterine pregnancy.

## 1. Background

Interstitial pregnancy (IP) is a rare ectopic pregnancy (EP), accounting for 2–4% of tubal EP [1]. The gestational sac is implanted into the interstitial part of the fallopian tube, and the surrounding blood supply is rich. Once ruptured, fatal bleeding may occur [2]. The maternal mortality rate is as high as 2.5%, which is seven times that of other types of tubal EP [3]. The coexistence of an IP and an intrauterine pregnancy (IUP) is called a heterotopic interstitial pregnancy (HIP), which is the most life-threatening type of EP. During natural conception, the incidence of HIP is very low [4]. In recent years, with the increase in infertility and the development of assisted reproductive technology (ART), the incidence of HIP has greatly increased [5]. The existence of IUP also increases the complexity of HIP diagnosis and treatment. Thus, attention must be paid to the following issues: the desire of the expectant parents for the health of the baby in utero and the protection of future fertility, and the threat to the life of the mother and intrauterine fetus by ectopic gestational sac rupture.

However, there is currently no unified treatment standard for HIP. We share a case of ultrasound-guided methotrexate intervention for HIP and review the literature on the treatment. Written consent for this case has been obtained and kept in the medical record.

## 2. Case Description

A 31-year-old woman came to Peking University Shenzhen Hospital complaining of slight vaginal bleeding and tenesmus for 8 h. She had undergone in vitro fertilization (IVF) and transferred two frozen embryos into the uterine cavity at the hospital 38 days prior. On the 14th day after embryo transfer (ET), she routinely received a serum β-human chorionic gonadotropin (β-hCG) test, which showed 2372 IU/L, indicating pregnancy. On day 30 after ET, the patient had her first ultrasound, which revealed an embryo in the uterus. During this period, the patient and the embryo were in a stable condition. However, on the 38th day after ET, she had a small amount of vaginal bleeding and tenesmus, so she returned to the hospital for treatment. Two gestational sacs were unexpectedly discovered by transvaginal ultrasonography (TVUS) on day 38 after ET: one was in the uterine cavity with a fetal bud of 15 mm long and with a fetal heartbeat, and the other was located in the right interstitial part of the fallopian tube, which was not communicated with the uterine cavity. In this gestational sac mass, there was a fetal bud about 5 mm long, and the fetal heart rate was not yet obvious, but the blood flow signal around the mass was abundant (Figure 1). In addition, the patient’s serum β-hCG level was shown to be higher than 264,000 IU/L. She was finally diagnosed with HIP based on TVUS, serum β-hCG levels, and multidisciplinary expert consultation.

Given that this IP was still growing, it was necessary to promptly inactivate the EP. Two treatment options were presented to the patient: laparoscopic surgery or fetal reduction treatment. Ultimately, the patient and her partner gave written consent for the patient to undergo the ultrasound-guided local injection of methotrexate (MTX) to inactivate the interstitial gestational sac after fully understanding the possible risks to the mother and fetus. For the procedure, the patient lay in the supine position with her lower abdomen fully exposed. The puncture point was chosen, and the drape was then routinely disinfected. Under the real-time guidance of ultrasound, an 18G puncture needle was advanced to the villus implantation site of the right IP, after which 20 mg of MTX (0.4–0.5 mg/kg body weight was dissolved in 5 mL of normal saline) was slowly injected into the interstitial gestational sac [6]. The needle was then pulled out after inserting the needle core. The process went smoothly. Subsequent serial ultrasound scans and serum β-hCG level tests were performed to dynamically observe the changes in the EP and IUP. On the second day after injection, the interstitial gestational sac measured about 32 × 22 mm, and there was a slight streak-like blood flow signal around it. On the 7th day, there was no significant change in the ectopic gestational sac, and the serum β-hCG level was 235,123 IU/L. On the 14th day, the ectopic gestational sac was larger, about 44 × 43 mm in size, with abundant peripheral blood vessels (Figure 2). On the 18th day, the ectopic gestational sac had grown to its largest size, about 58 × 40 mm. After this, it became increasingly smaller until the 22nd week of gestation, when it was about 35 × 30 mm in size and surrounded by fewer blood vessels. On the 20th day, the serum β-hCG level of the patient was 85,702 IU/L. The changes in the size of the interstitial gestational sac and the serum β-hCG level are respectively presented in Figure 3A,B. 

The patient had regularly been undergoing prenatal examination in the hospital until the 22nd week of gestation. After that, she visited another hospital, and was followed up with via telephone. The IUP evolved normally and uneventfully. The patient gave birth to a boy with patent ductus arteriosus (PDA) in another hospital at 38 weeks of pregnancy. The interstitial gestational sac finally disappeared on the 51st day after delivery.

## 3. Literature Review

### Materials and Methods

The search terms “heterotopic interstitial pregnancy [all fields]”, “((interstitial pregnancy) AND heterotopic pregnancy [all fields])”, or “interstitial pregnancy complicated with intrauterine pregnancy [all fields]” were used to search studies published on the PubMed database in English from 1992 to 2021, including references and review articles of relevant case reports. Additionally, reports of angular heterotopic pregnancy were excluded.

All case reports of HIP were selected, regardless of the treatment or outcome, and reported data on clinical manifestations, risk factors, treatments, and outcomes were registered. Some reports did not contain all of these data. 

## 4. Result

Based on the search strategy described previously, 120 relevant English documents published on PubMed were retrieved. After reading the full text, 24 articles with 25 cases (Table 1) were finally selected. Including the present case, a total of 26 cases were included.

It was found that patients in 84.6% (22/26) of these cases had conceived through in vitro fertilization embryo transfer (IVF-ET). Moreover, 57.7% (15/26) of the patients in these cases had fallopian tube disorders, and 23.1% (6/26) had a history of EP.

Regarding clinical manifestations, 53.8% (14/26) of the patients in these cases complained of abdominal pain, 19.2% (5/26) had symptoms of vaginal bleeding, and 26.9% (7/26) had no clinical symptoms. All cases were confirmed by TVUS. In these cases, the earliest diagnosis of HIP was 21 days after ET. The IP ruptured in 46.2% (12/26) of the cases, all of which were treated with surgery. Furthermore, 53.8% (14/26) of the patients in these cases did not experience IP rupture, of which 78.6% (11/14) were treated with ultrasound-guided local drug injection. Regarding the drugs used for local injection, potassium chloride (KCl) was used in four cases, MTX was used in three cases, KCl and MTX were used in one case, MTX and dactinomycin were used in one case, and hypertonic glucose was used in two cases. In addition, two patients with an unruptured interstitial gestational sac underwent laparotomy, and one patient with an unruptured interstitial gestational sac underwent laparoscopic surgery.

In total, 76.9% (20/26) of the reported IUPs had a good outcome, including 10 cases of surgical treatment and 10 cases of ultrasound-guided local drug injection. All newborns were healthy. On the contrary, the poor outcomes of IUP included one fetus with trisomy 21, which resulted in the spontaneous abortion of the IUP. Moreover, patients in three cases had been diagnosed with IUP discontinuation before receiving treatment, and two patients had a postoperative IUP abortion.

## 5. Discussion

High-risk factors for heterotopic pregnancy (HP) include a history of unilateral or bilateral salpingectomy [11,13,16,18,23,24,28,29,30], EP [11,16,18,24,28,29], ovulation induction, and IVF-ET [8,9,10,11,12,13,14,15,16,18,20,21,22,23,24,25,26,27,28,29,30,31,32]. Particularly, IVF-ET increases the occurrence of IP [33]. EP mostly occurs after fresh embryo transfer and multiple embryo transfer, while the incidence of EP after cleavage-stage transfer and blastocyst-stage transfer is basically similar [34]. Therefore, frozen single blastocyst transfer is now advocated [32].

The diagnosis of HIP mainly relies on ultrasonography, especially TVUS [35]. It has been reported that the sensitivity and specificity of TVUS for HP are 92.4% and 100%, respectively, and the positive and negative predictive values are 100% and 99.9%, respectively [36]. Ultrasonographic criteria for diagnosing an EP mass as an IP are as follows: (1) an eccentric gestational sac (at least 1 cm from the outermost point of the endometrium); (2) the thinning (<5 mm) of the surrounding superficial myometrium; (3) the interstitial line sign, which represents the interstitial part of the tube or the endometrial tube extending from the corner to the middle of the interstitial mass [37]. Our case was diagnosed as an IP based on the sonographic appearance of an ectopic gestational sac that was not connected to the endometrium and was surrounded by a thin interrupted myometrium. In fact, post-embryo transfer luteinizing cysts and ovarian hyperstimulation often lead to atypical ultrasound images. Therefore, medical staff should be highly vigilant for indirect ultrasound signs such as adnexal mass and pelvic effusion. If ultrasound cannot identify the location of the EP, MRI may be a further option. However, due to the presence of IUP, serum β-hCG levels have a limited role in diagnosing HP. 

The treatment of HIP aims to terminate IPs, maintain IUPs, and achieve good pregnancy outcomes. In this study, the survival rate of early intervention with IUP was found to reach 76.9%, which is basically consistent with the value of 83.3% reported in the literature [38]. Commonly used clinical treatment methods include surgery, ultrasound interventional therapy, and expectant therapy. Surgery is the main treatment option for ruptured EP [37]. Keratostomy and corneal wedge resection are now more commonly used methods [39]. Surgery can quickly and definitively remove an EP mass, and the IUP live birth rate can reach 80–84% [32]. However, there is also a risk of persistent EP [39] and intrauterine growth retardation [40]. In addition, the operative time tends to be longer, with an average of 55.5 ± 21.3 min, and the average intraoperative blood loss is 76.0 ± 73.2 mL [41]. Surgery also increases the risk of uterine rupture in another pregnancy by about 33% [42]. The effect of surgery on future fertility is controversial. However, recent studies have shown that laparoscopic electrosurgery and mechanical traction do not impair ovarian reserve, and the concentration of anti-Müllerian hormone in the first trimester of re-pregnancy is normal [43]. Asgari et al. [44] followed 194 patients with tubal EP for 18 months and found no significant difference in fertility after MTX, salpingostomy, and salpingectomy. A systematic study showed that salpingectomy for EP had no negative effect on ovarian reserve and ovarian response [45]. From another retrospective study, we also found no difference between the re-pregnancy rates at 3 years of patients undergoing laparoscopic salpingectomy and salpingostomy for their first EP [46]. Surgery also seems to work like medicine without having effects on fertility [47].

Ultrasound interventional therapy refers to ultrasound-guided local drug injection to inactivate the ectopic gestational sac, which is a minimally invasive treatment method that can maximize the integrity of the uterus and reduce the risk of future uterine rupture [8]. The successful delivery rate of IUP after interventional ultrasound can reach 100% [6]. Available drugs are KCl [7,8,14,21] MTX [15,18,29] and hypertonic glucose [20]. Since the first report of the successful treatment of HIP by the ultrasound-guided injection of KCl in 1992 [7], many medical centers have carried out studies, all of which have achieved good results. However, KCL mainly acts on embryo or fetal heart activity, which leads to fetal death, and does not inhibit trophoblast proliferation [48]. As a result, self-limited vaginal bleeding has occasionally been reported during the duration of an IUP [49]. To overcome the inability of KCl to inhibit trophoblast proliferation, Baker et al. [11] combined MTX and KCl to inactivate IP. In 1982, Tanaka et al. [50] reported the intramuscular injection of MTX in the treatment of IP. To the best of our knowledge, this was the first case that maintained tubal patency and protected reproductive function. To date, conservative treatment with multiple doses and/or topical MTX has become the standard of care for IP; however, systemic MTX is contraindicated in the presence of IUP [6,51,52] because high doses of MTX may cause fetal congenital bone deformity or IUP miscarriage [53]. Nevertheless, low-dose MTX treatment has been shown to have no effect on coexisting IUP [29]. The ultrasound-guided local injection of 25 mg MTX has been reported to terminate IPs without affecting IUPs [54,55]. Studies have also proved that MTX doses within the range of 12.5–30 mg will not affect the growth and intelligence of infants [6]. In our case, the patient was diagnosed with IP in the first trimester, and had stable vital signs and no EP rupture. This case was suitable for ultrasound interventional therapy, which can also satisfy the strong willingness of patients to protect IUPs. Therefore, we treated the patient with the local injection of a low dose (20 mg) of MTX under ultrasound guidance. The short-term enlargement of the ectopic mass occurred on the 7th day after the operation, which may have been due to the transient change in the ectopic mass tissue without villous activity as gestation proceeded after interventional treatment. Dynamic ultrasound monitoring revealed that the ectopic mass eventually disappeared. The outcome of this IUP was good. Although the infant had PDA, there have been no reports on the correlation between PDA and MTX. The prognosis of this patient will continue to be monitored, and the health and development of the baby will be followed up with until his childhood to record any unexpected problems that may have been related to MTX exposure in the uterus. This therapy is suitable for patients with stable vital signs, a small gestational age, an unruptured ectopic gestational sac, or contraindications to laparoscopic surgery. Moreover, it is less expensive, and the recovery period is shorter. 

In addition, expectant therapy is another possible treatment modality. It is mainly suitable for patients with ectopic gestational sac atrophy, stable vital signs, and close follow-up. Wu et al. [56] reported seven cases of suspected HP with an EP mass less than 4 cm in length, and they achieved good outcomes after expectant treatment. Certainly, there are other reports of IUP miscarriage (1/3) [55]. However, based on the current limited clinical research [57,58,59], the survival rate of IUP with this treatment remains unclear. Thus, it is essential to dynamically evaluate the safety of this therapy when choosing it, and surgical treatment should be applied if necessary.

In short, the diagnosis and treatment of HIP remain challenging. Doctors must establish a complete medical record, carefully inquire about the patient’s past medical history and family history, and repeat inspections for suspicious patients or assisted-pregnancy patients to avoid a missed diagnosis. A systematic examination of the bilateral adnexal uterus cannot be ignored even if the presence of IUP is confirmed. In addition, patient complaints should be noted. Not all patients with HP experience symptoms such as abdominal pain, vaginal bleeding, or tenesmus; back pain [4], gastrointestinal symptoms [38], or pelvic pain [60] may sometimes also be suggestive of HP. Moreover, routine TVUS and regular follow-up after IVF-ET are conducive to the early diagnosis of HP. The treatment of HP must be considered in combination with the patient’s clinical manifestations and needs, and the physician’s experience. 

## Figures and Tables

**Figure 1 jpm-13-00332-f001:**
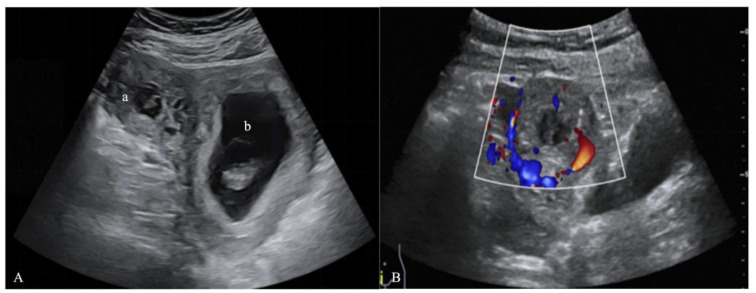
Ultrasound images on the 38th day after ET. (**A**) A two-dimensional ultrasound image: (a) the interstitial gestational sac in the interstitium of the right fallopian tube and (b) the intrauterine gestational sac. (**B**) A color Doppler ultrasound image showing an abundant blood flow signal around the IP.

**Figure 2 jpm-13-00332-f002:**
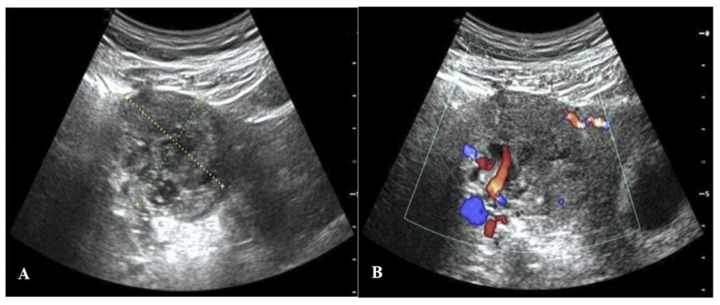
Ultrasound images on the 14th day after the injection of MTX. (**A**) A two-dimensional ultrasound image, which reveals the growth of the ectopic gestational sac. (**B**) A color Doppler ultrasound image showing an abundant blood flow signal around the IP.

**Figure 3 jpm-13-00332-f003:**
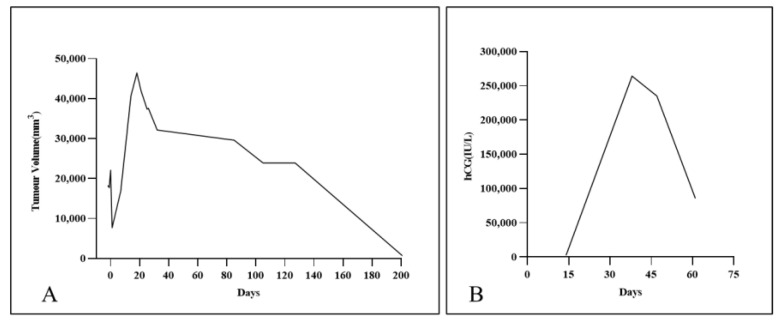
(**A**) The change in the size of the interstitial gestational sac. (**B**) The change in the serum β-hCG level. The day of injection was set as day 0. Mass size = long diameter short diameter^2^/2.

**Table 1 jpm-13-00332-t001:** Summary and characteristics of relevant literatures and cases.

Reference	Age (Years)	Medical History	Manner of Fertilization (Embryo Type and Number if ET)	Symptoms	Site of TP	Time of Admission *	EP Rupture (Yes or NO) + Intervention	IP Outcome	IUP Outcome + Neonatal Malformation (Yes or No)
Leach et al., 1992 [7]	29	Bilateral isthmic-isthmic tubal anastomosis	Intrauterine insemination	PP + VB	IUP + right IP	8 weeks	No + KCL	Inactivated	CS + No
Pérez et al., 1993 [8]	35	Secondary tubal infertility	IVF-ET (fresh 4)	-	IUP + left IP	33 days after ET	No + KCL	Inactivated	FC + No
Sherer et al., 1995 [9]	32	-	IVF-ET (6)	ABP	IUP + IP	8 weeks	Yes + LS	Inactivated	CS/33 weeks + No
Luckas et al., 1997 [10]	24	Secondary infertility due to severe tubal damage	IVF-ET (fresh 3)	ABP + UT	IUP + right IP	8 weeks	Yes + LT	CS/30 weeks+ newborn health	CS/30 weeks + No
Baker et al., 1997 [11]	34	Left ST + EP	IVF-ET (fresh 4)	-	IUP + IP	34 days after ET	No + KCL and MTX	Inactivated	FC + No
Dumesic et al., 2001 [12]	37	BH	IVF-ET (4)	ABP	IUP + IP	4 weeks	Yes + LS + LT	Inactivated	SA
Nikolaou et al., 2002 [13]	31	Left oophorectomy + partial ST	IVF-ET (fresh 2)	Intermittent discomfort in the left iliac fossa	IUP + left IP	8 weeks	Yes + LT	Inactivated	ND/33 weeks + No
Ghazeeri et al., 2002 [14]	29	Endometriosis	IVF-ET (3)	-	IUP + right IP	28 days after ET	No + KCL	Inactivated	ND/36 weeks + No
Oyawoye et al., 2003 [15]	29	PI	IVF-ET (3)	-	IUP + left IP	6 weeks + 3 days	No + MTX	Inactivated	ND/39 weeks + No
Chang et al., 2003 [16]	21	Bilateral ST + EP	IVF-ET (fresh 3)	VB	IUP + left IP	35 days after ET	No + LT	Inactivated	CS/38 weeks + No
Lialios et al., 2008 [17]	29	-	SC	ABP	IUP + right IP	6 weeks + 5 days	Yes + LS	Inactivated	CS/37 weeks + No
Fujioka et al., 2008 [18]	39	Left ST + EP	IVF-ET (frozen 2)	VB	Intrauterine blighted ovum pregnancy + right IP	28 days after ET	No + MTX IM, 3 times, 50 mg/m^2^ each time + dactinomycin 12 mg/kg	Inactivated	SA
Thomas et al., 2011 [19]	-	-	-	PP + VB	IUP + right IP	6 weeks	No + LS	Inactivated	SA
Wang et al., 2012 [20]	29	PI	IVF-ET (frozen 3)	-	IUP + right IP	21 days after ET	No + IP aspiration and hypertonic glucose infusion guided by ultrasound guidance	Inactivated	CS/35 weeks + No
Wang et al., 2012 [20]	27	PI	IVF-ET (fresh 2)	-	IUP + right IP	30 days after ET	No + IP aspiration and hypertonic glucose infusion guided by ultrasound guidance	Inactivated	ND/37 weeks + No
Savelli et al., 2014 [21]	27	-	IVF-ET (2)	-	IUP + right IP	6 weeks + 5 days	No + KCL	Inactivated	CS/35 weeks + No
Xu et al., 2015 [22]	28	PI + bilateral tubal ligation	IVF-ET (fresh 2)	ABP	IUP + left IP	48 days after ET	Yes + LT	Inactivated	SA
Paradise et al., 2016 [23]	38	FTS	IVF-ET (-)	ABP	IUP + IP	26 weeks +	Yes + LT	CS/26 weeks+ newborn health	CS/26 weeks + No
Aoki et al., 2016 [24]	33	Left ST+ EP	IVF-ET (frozen 1)	ABP	IUP + IP	28 days after ET	Yes + LS	Inactivated	CS/38 weeks + No
Dendas et al., 2017 [25]	35	Right accessory resection + laparoscopic oophoroplasty and adhesion lysis	IVF-ET (frozen 2)	Intermittent colicky pain in the right iliac fossa with nausea and constipation	IUP + IP	16 weeks	Yes + LS + LT	Inactivated	CS/33 weeks + No
Chen et al., 2017 [26]	27	Bilateral tubal obstruction with hydrosalpinx	IVF-ET (frozen 3)	ABP +VB	IUP + right IP	31 days after ET	Yes + LS	Inactivated	ND/39 weeks + No
Anderson et al., 2018 [27]	27	-	-	ABP	IUP + left IP	9 weeks	Yes + LT	Inactivated	SA
Siristatidis et al., 2018 [28]	35	FTS + EP hydrosalpinx	IVF-ET (frozen)	ABP	Intrauterine withered egg pregnancy + right IP	8 weeks + 3 days	No + LT	Inactivated	SA
Lee et al., 2020 [29]	34	Cervical cancer + Left FTS + EP	IVF-ET (-)	-	IUP + left IP	8 weeks	No + transabdominal cerclage+ MTX	Inactivated	CS/38 weeks + No
Wang et al., 2021 [30]	29	FTS	IVF-ET (frozen 2)	ABP	IUP + left IP	5 weeks	Yes + LS	Inactivated	CS/39 weeks + No
Present case	31	PI	IVF-ET (frozen 2)	VB	IUP + Right IP	8 weeks	Yes + MTX	Inactivated	ND/39 weeks + PDA

Notes: ET = embryo transfer, LS = laparoscopy, LT = laparotomy, ABP = abdominal pain, VB = vaginal bleeding, PP = pelvic pain, UT = uterine tenderness, KCL = KCL injection guided by ultrasound guidance, MTX = MTX injection guided by ultrasound guidance, cause for salpingectomy bilateral ectopic pregnancies, BH = bilateral hydrosalpinx, FTS = fallopian tube resection, ST = bilateral salpingectomy, PI = primary infertility, CS = cesarean section, FC = full-term childbirth, SA = spontaneous abortion, SC = spontaneous conception, ND = natural delivery. * Refers to the time after embryo transfer.

## Data Availability

Data are available on request from the authors.

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
