# Peer review of "Successful Ultrasound-Guided Methotrexate Intervention in the Treatment of Heterotopic Interstitial Pregnancy: A Case Report and Literature Review"

_jpm, 2023, doi:10.3390/jpm13020332_

Round 1

Reviewer 1 Report

The manuscript is a case report and literature review on successful ultrasound intervention in the treatment of heterotopic interstitial pregnancy (HIP). 

The authors follow in general terms the PRISMA-IPD guidelines reporting the diagnosis and successful treatment of a patient with HIP for whom transvaginal ultrasound (TVUS) played a primordial role in the required delivery of care.

The background with the patient’s case description is well documented. It is of utmost importance to revise Figure 1 (A) and make sure that (a) and (b) are correctly labeled. (a) seems to correspond to the interstitial gestational sac, while (b) seems to correspond to the intrauterine gestational sac.

For the literature review materials and methods, the authors clearly describe the methods for identifying the selected studies, including the search terms and inclusion criteria for which Table 1 provides a concise and clear summary. 

The Discussion section states high-risk factors for heterotopic pregnancy (HP) and current recommendations for blastocyst transfers; sensitivity and specificity of TVUS for HP; surgery, expectant therapy; and the effects of such methods of care on the patients and fetuses or infants. The ultrasound interventional therapy is clearly explained in more detail, regarding its procedure and special attention is paid to its application to the patient particularly being discussed.

The authors conclude their manuscript summarizing some of the challenges that remain in the diagnosis and treatment of HIP together with their recommendations.

Author Response

Dear  Reviewers,

We would like to express our sincere appreciation for your careful reading and invaluable comments to improve this manuscript. We have addressed all issues raised by the reviewers. The amendments made are mentioned below with reference to appropriate paragraphs and sections of the revised manuscript.

Regarding the error labels (a) and (b) in Figure 1 (a), we have modified their comments respectively (P13L12-14 ).

Thank you again for your comments and we look forward to hearing from you regarding our submission. We would be glad to respond to any further questions and comments that you may have.

Reviewer 2 Report

Review of an article:

Successful ultrasound intervention in the treatment of heterotopic interstitial pregnancy: A case report and review of the literature

Summary of the article:

This case report a literature review is well written and performed.  It contains a case report of treatment of heterotopic interstitial pregnancy by ultrasound-guided methotrexate injection. The systematic review summarizes literature on treatment of  heterotopic interstitial pregnancy.

Comments:

1/ I do suggest change of the title (as the,, ultrasound intervention,, sound misleading the intervention is just guided by ultrasound, the intervention is injection of methotrexate

i.e. Successful ultrasound-guided methotrexate intervention in the treatment of heterotopic interstitial pregnancy: A case report and literature review  

this terminology I suggest to keep through all article

2/ abstract

 Early treatment of intrauterine pregnancy is associated with a high survival rate.  – please rather rephrase

i.e. I guess you mean:  Early treatment of concomitant heterotopic pregnancy is associated with high survival of the intrauterine pregnancy.

3/ Intro

 Thus, attention must be paid to the following two issues: the desire of the expectant parents for the 1/ health of the baby in utero and 2/ the protection of future fertility, and 3/ the threat to the life of the mother and intrauterine fetus by ectopic gestational sac rupture  -  I see three issues

4/ methods

. In these cases, the earliest time taken to diagnose HIP varied from 21 days to 30 weeks after ET – really 30 weeks?

5/  The IP ruptured in 46.2% (12/26) of the cases, all of which were treated with laparoscopic surgery or laparotomy. – please try to rephrase, that it is easier to read  – that it is clear that those 26 were treated surgically

6/ I think in the article should be somewhere more clearly stated, that MTX is teratogenic and therefore the use is still controversial if i.e. KCl can be used

i.e. https://www.ajog.org/article/S0002-9378(98)70654-4/fulltext

I was pleased to review this article

Author Response

Dear Editors and Reviewers,

We would like to express our sincere appreciation for your careful reading and invaluable comments to improve this manuscript. We have addressed all issues raised by the reviewers. The amendments made are mentioned below with reference to appropriate paragraphs and sections of the revised manuscript.

1、We have revised the title of the article and the corresponding part of the article. (P1L1-2, 5-6,22-23; P2L19 ).

2、As for the "abstract" part, we have modified it in article P1L26-27.

3、We have corrected the "Intro" problem in article P2L15.

4、We have corrected the "methods" problem in article P8L12-13 and the corresponding part of table 1.

5、As for the fifth question, we have revised it in article P8L14.

6、About methotrexate and KCL, we have explained in part P11L8-11, 18-22.

Thank you again for your comments and we look forward to hearing from you regarding our submission. We would be glad to respond to any further questions and comments that you may have.
